# Lantox—The Chinese Botulinum Toxin Drug—Complete English Bibliography and Comprehensive Formalised Literature Review

**DOI:** 10.3390/toxins13060370

**Published:** 2021-05-22

**Authors:** Dirk Dressler, Lizhen Pan, Junhui Su, Fei Teng, Lingjing Jin

**Affiliations:** 1Movement Disorders Section, Department of Neurology, Hannover Medical School, Carl-Neuberg-Str. 1, 30625 Hannover, Germany; 2Neurotoxin Research Center of Key Laboratory of Spine and Spinal Cord Injury Repair and Regeneration of Ministry of Education, Tongji University School of Medicine, 389 Xincun Road, Shanghai 200065, China; jasonsu@tongji.edu.cn (J.S.); tengfei@tongji.edu.cn (F.T.); lingjingjin@tongji.edu.cn (L.J.); 3Department of Neurology, Shanghai Tongji Hospital, Tongji University School of Medicine, 389 Xincun Road, Shanghai 200065, China; 4Department of Neurorehabilitation, Yangzhi Rehabilitation Hospital (Shanghai Sunshine Rehabilitation Center), Tongji Univeirsity School of Medicine, No. 2209 Guangxing Rd, Shanghai 201619, China

**Keywords:** botulinum toxin, therapy, Chinese botulinum toxin, lanbotulinumtoxinA, formalised review, complete English bibliography

## Abstract

In 1997, lanbotulinumtoxinA (LAN) was introduced in China. It is now available in Asia, Latin America and Eastern Europe under various brand names including Hengli^®^, Lantox^®^, Prosigne^®^, Lanzox^®^, Redux^®^, Liftox^®^, HBTX-A and CBTX-A. The literature on LAN is mostly published in Chinese language, restricting its international accessibility. We, therefore, wanted to generate a complete English bibliography of all LAN publications and then use it for a comprehensive formalised literature review. Altogether, 379 LAN publications (322 in Chinese and 57 in English) were retrieved from PubMed and Science and Technology Paper Citation Database. Indications covered are motor (257), glandular (16), pain (32) and aesthetics (48). Topics are neurological (250), aesthetic (48), paediatric (38), ophthalmological (18), urological (9), methodological (6), gastroenterological (5), ear, nose and throat (4) and surgical (1). Seventy-one publications are randomised controlled trials, forty-one publications are interventional studies and observational studies, fifteen publications are case studies, eighteen publications are reviews, and two publications are guidelines. LAN publications cover all relevant topics of BT therapy throughout a period of more than 20 years. This constitutes a publication basis resembling those of other BT drugs. None of the LAN publications presents data contradictory to those generated with other BT type-A drugs. LAN seems to have a similar efficacy and safety features when compared to onabotulinumtoxinA using a 1:1 LAN– onabotulinumtoxinA conversion ratio. Large controlled multicentre studies will become necessary for LAN’s registrations in Europe and North America.

## 1. Introduction

Botulinum toxin (BT) is produced by the anaerobic bacterium *Clostridium botulinum*. In the early 19th century, Justinus Kerner first recognised BT as the cause of botulism. In 1980, Alan B. Scott invented BT’s therapeutic use by applying it as a muscle relaxant for the treatment of strabismus in children [1]. In the meantime, BT’s use has expanded tremendously into numerous therapeutic indications and into widespread aesthetic use.

Globally, there are three major BT drugs available: onabotulinumtoxinA (ONA, Botox^®^, A; Abbvie, Chicago, IL, USA), abobotulinumtoxinA (ABO, Dysport^®^; Beaufour Ipsen, Boulogne-Bilancourt, France) and incobotulinumtoxinA (INCO, Xeomin^®^; Merz Pharmaceuticals GmbH, Frankfurt/M, Germany) [2]. They are all based on BT type A and therefore have similar efficacy and adverse effect profiles. Whilst the potency labelling of ONA and INCO is identical, the potency labelling of ABO is different and conversion ratios to compare ABO with other BT drugs are still a matter of debate.

In China, the development of therapeutic BT started in 1985 by Professor Yinchun Wang at the Lanzhou Institute of Biological Products. Wang was trained by Edward J. Schantz at the University of Wisconsin in botulism research. In 1993, his product was approved by the Ministry of Health of the People’s Republic of China. This made China—after the USA and the UK—the third country in the world producing therapeutic BT. In 1997, the Lanzhou product was licensed by the Chinese Food and Drug Administration under the name lanbotulinumtoxinA (LAN, Lantox^®^; Lanzhou Institute of Biological Products, Lanzhou, China) as a drug for the treatment of strabismus, blepharospasm and hemifacial spasm. In 2012, this license was extended to moderate to severe glabellar frown lines. In 2002, LAN was successfully registered in South Korea. Since then, it has spread over Asia, Latin America and Eastern Europe, including major countries such as Brazil, India, Mexico and Russia [3]. LAN is available under various brand names such as Hengli^®^, Prosigne^®^, Lantox^®^, Lazox^®^, Redux^®^, Liftox^®^, HBTX-A and CBTX-A. In this review, we will use LAN as the abbreviation of the generic name lanbotulinumtoxinA.

BT drugs consist of botulinum neurotoxin (BNT) as their active pharmaceutical ingredient, complexing proteins and excipients [4]. In LAN, BNT has a molecular weight of 150 kDa with a heavy chain of 100 kDa and a light chain of 50 kDa. Its chemical purity is more than 99.5% [5]. LAN’s complexing proteins include the hemagglutinin (HA) components HA70 (molecular weigts: 57 kDa and 17 kDa), HA33 (molecular weigts: 30 kDa and 28 kDa) and HA17 (molecular weigt: 15 kDa) and the non-toxic non-hemagglutinin component NTNH (molecular weigt: 136 kDa) [6]. These basic components are similar to ONA and ABO. However, the manufacturing process differs and the stabilising protein used in LAN is not human serum albumin, but bovine gelatin 5 mg per vial. Other excipients include dextran of 25 mg per vial and sucrose of 25 mg per vial. The role of gelatin as well as LAN’s potency labelling will be discussed below. BT’s clinical pharmacology was recently reviewed in detail [7].

The literature on LAN is mostly published in Chinese language and in Chinese journals restricting its international accessibility. We, therefore, retrieved the entire LAN literature from Chinese and English databases. The English translations of the publications allowed us to generate a complete English bibliography and to perform a formalised review using commonly accepted classification criteria.

## 2. Results

### 2.1. General

As shown in Table 1, altogether 379 LAN publications were retrieved: 322 publications are in Chinese, and 57 publications are in English. Five of the fifty-seven English publications appear in Chinese journals, and fifty-two publications come out of non-Chinese sources. Publication languages other than English and Chinese were not retrieved.

Two-hundred and fifty-seven publications cover motor indications, sixteen publications include glandular indications, thirty-two publications include pain, forty-eight publications include aesthetic indications, six publications include methodological aspects, and twenty publications are reviews and guidelines.

Two-hundred and fifty publications are research in the neurological field, forty-eight publications are in the aesthetic field, thirty-eight publications are in the paediatric field, eighteen publications are in ophthalmologic field, nine publications are about urologic indications, five publications are about gastroenterologic indications, four publications are regarding ear, nose and throat, and one publication is regarding surgical indications.

Two-hundred and thirty-one publications are about observational studies, seventy-one publications are about randomised controlled trials, forty-one publications are about interventional studies, eighteen publications are reviews, fifteen publications are case studies, and two publications are guidelines.

In the following, we will summarise the number of publications and discuss in detail some selected randomised controlled trials, interventional studies, reviews and guidelines for each indication.

### 2.2. Motor Indications

Motor indications are indications where the therapeutic effect is based on the blockade of the cholinergic innervation of hyperactive muscle tissue. Those are the first indications of BT therapy. Whether the entire therapeutic effect relies on a direct peripheral effect alone is not entirely clear. Retrograde axonal transport [8] may also affect spinal or higher central nervous system structures [9]. Additionally, indirect effects on the central nervous system may also contribute to BT’s therapeutic effect.

#### 2.2.1. Dystonia

Dystonia was—after strabismus and together with hemifacial spasm—the second group of indications developed. It is still the largest indication group for BT’s therapeutic use. Altogether, 106 LAN publications refer to the treatment of dystonia. In most publications, more than one dystonia manifestations are covered. Publications on blepharospasm also include the treatment of Meige syndrome, hemifacial spasm and sometimes cervical dystonia.

Thirty-five publications (thirty-two as primary topic publications and three as secondary topic publication) cover blepharospasm. They deal with the efficacy and safety of LAN therapy, injection sites, injection patterns and its comparison to that of surgical treatment. Sometimes, large cohorts are reported, and long-term experience is shared. Table 2 shows two randomised controlled trials on LAN in blepharospasm. One of them [10] showed no difference in efficacy and safety between LAN and ONA. The other one [11] showed that sparing the medial lower eyelid improves tear film stability and lacrimal fluid drainage. One cross-over interventional study [12] found no difference in efficacy and safety between LAN and ONA. All comparisons were based on a dose conversion ratio between LAN and ONA of 1:1. Seventeen publications (thirteen as primary topic publicationss and four as secondary topic publications) cover Meige syndrome. They deal with the efficacy and safety of LAN therapy, combinations with spasmolytic drugs and combinations with wrinkle treatment. Sometimes, large cohorts are reported, and long-term experience is shared. Most studies also include patients with blepharospasm and hemifacial spasm.

Thirty-two publications (all as primary topic publications) cover cervical dystonia. They deal with injection guidance by electromyography or ultrasound, variations of LAN concentrations, drug comparisons between LAN, ONA and ABO, combinations of LAN and orthoses, combinations of LAN and thermocoagulation of peripheral nerves, the treatment of tardive cervical dystonia, as well as long-term studies and reports on larger cohorts. Table 2 shows six selected randomised controlled trials for cervical dystonia. One study [13] compared LAN and ONA at a 1:1 dose conversion ratio, and one study [3] compared LAN and ABO at a 1:3 dose conversion ratio. Both studies did not find a difference in efficacy and safety between both drugs. One study [14] showed that LAN together with orthopaedic joint braces has a better efficacy than LAN alone, whereas oral drugs are not effective. One study [15] showed that electromyographic guidance produces identical efficacy and less adverse effects but was more painful. Therapeutic effects are also longer. Two studies [16,17] demonstrated that different LAN dilutions do not affect efficacy and safety.

Five publications (all as primary topic publications) cover craniocervical dystonia, two publications are about writer’s cramp (all as primary topic publications), tweleve publications are about dystonias (eleven as primary topic publications and one as a secondary topic publication), and three are about spasmodic dysphonia (all as primary topic publications). They mainly report larger cohorts, long-term follow-ups and general efficacy and safety data. Table 2 shows the randomised controlled trial on adductor-type spasmodic dysphonia [18]. It was demonstrated that LAN with a dose of 10 MU produces longer effects than LAN with dose of 5 MU whereas peak efficacy and safety of these two doses are not different.

Three publications refer to the treatment of temporomandibular joint dysfunction, with all as primary topics. The aetiology of the condition is mixed (habitual dislocation, masticatory spasms and masticatory pain), and potential therapeutic mechanisms remain unclear. Table 2 shows the randomised controlled trial on temporomandibular joint disorders [19]. It was demonstrated that semiconductor laser therapy with LAN has better efficacy on inflammation and pain than semiconductor laser therapy alone.

#### 2.2.2. Hemifacial Spasm

Hemifacial spasm is together with dystonia the second indication developed for BT therapy following strabismus. As shown in Table 1, 35 LAN publications refer to the treatment of hemifacial spasm (17 as primary topic publications and 18 as secondary topic publications). Table 2 shows two randomised controlled trials and two interventional studies on LAN for hemifacial spasm, one randomised controlled trial [10] and one interventional study [12] by comparing LAN and ONA, and no difference in efficacy and safety was found when converted on a LAN and ONA dose conversion ratio of 1:1. One randomised controlled trial [20] showed additional posterior auricular muscle injections can reduce acoustic symptoms, and one interventional study [21] showed that the LAN dose of 50 MU/mL produces longer therapeutic effects and more adverse effects than that of 25 MU/mL whereas the peak efficacies at these two doses are identical.

#### 2.2.3. Tics

Tics have been tried successfully on several occasions, because of their similarity with dystonia, especially cranial dystonia. As shown in Table 1, one LAN publication (as a primary topic publication) refers to tics and is an observational study.

#### 2.2.4. Spasticity

Spasticity is another main indication for BT therapy. Historically, it was developed after strabismus, dystonia and hemifacial spasm. As shown in Table 1, 40 LAN publications (all as primary topic publications) refer to spasticity. Apart from general data on efficacy and safety, they cover different aetiologies (stroke, spinal cord injury and brain damage), combinations with other methods (rehabilitation training, physiotherapy, acupuncture, Shujin Huoluo lotion—a traditional Chinese medicine to reduce spasticity and electric stimulation), different LAN doses, different LAN concentrations, different BT drugs, different spasticity locations (arms and legs) and injection guidance with ultrasound and electromyography.

As shown in Table 3, 10 randomised controlled trials deal with spasticity. Five of them are about arm spasticity, and five are about leg spasticity. One randomised controlled trial [22] demonstrated the general efficacy and safety of LAN against placebo. Four randomised controlled trials [23,24,25,26] demonstrated that LAN improves rehabilitation. One randomised controlled trial demonstrated that LAN improves acupuncture [27], and two randomised controlled trials reported that ankle brace [28] and spasmodic muscle training [29] improve LAN therapy. One randomised controlled trial [30] showed that intensified physiotherapy improves LAN therapy compared with regular physiotherapy, and one randomised controlled trial [31] suggested that using higher LAN doses (i.e., 400 MU) with lower concentrations (i.e., 50 MU/mL) may improve LAN therapy outcome.

#### 2.2.5. Cerebral Palsy

Cerebral palsy is a syndrome consisting of various neurologic deficits all caused by perinatal brain damage. Motor deficits are the hallmark, frequently including dystonic and spastic elements which have been treated successfully with BT therapy for many years. Economically, cerebral palsy turns out to be one of the most important BT indications, not only because of its prevalence, but also because of the usually high BT doses applied. As shown in Table 1, 38 LAN publications refer to the treatment of cerebral palsy, all as primary topics. They cover general efficacy and safety, prognosis of cerebral palsy under LAN therapy, dosing issues, co-therapies with serial casting and herbal baths, application guidance with electric stimulation and ultrasound, special or individual target muscles (arms, iliopsoas and triceps surae) and long-term follow-ups.

Table 4 shows six randomised controlled trials dealing with cerebral palsy. Two randomised controlled trials showed that LAN improves rehabilitation [32,33], and one randomised controlled trial showed that rehabilitation improves LAN therapy [34]. One randomised controlled trial suggested that LAN with a 6 MU/kg body weight has a stronger effect in more severe cerebral palsy [35], whereas one randomised controlled trial [36] found that LAN with different body weights (i.e., 3, 4 or 5 MU/kg) does not produce different effects. One randomised controlled trial showed that BT injection combining traditional Chinese medicine and Ueda, a physical therapy method for spasticity, could improve CPs more effectively, especially in motor function and balance ability [37].

#### 2.2.6. Strabismus

Strabismus is the first indication developed by Alan B. Scott for BT therapy. Its relevance today is limited. As shown in Table 1, 18 LAN publications refer to the treatment of strabismus, with 17 as primary topic publications and 1 as a secondary topic publication. The topics covered include general data on efficacy and safety, application together with hyaluronate, application guidance by ultrasound or electromyography, different aetiologies (idiopathic, intermittent exotropia, concomitant esotropia, thyroid ophthalmopathy and cranial nerve palsy) and comparison with surgery. One randomised controlled trial [38] showed that LAN injections with sodium hyaluronate have identical efficacy as LAN injections alone, but produce less ptosis.

#### 2.2.7. Bladder Dysfunction

BT therapy for bladder disorders is one of the established indications for BT therapy. Altogether, eight LAN publications refer to bladder dysfunction (all as primary topic publications). They cover different aetiologies (spinal cord injury and idiopathic), different LAN doses and different BT drugs. Different terminologies are used, such as overactive bladder, detrusor hyperreflexia, interstitial cystitis, bladder pain syndrome and neurogenic incontinence. Table 5 shows three randomised controlled trials on bladder dysfunctons. One of them [39] showed that LAN is more effective in female overactive bladder than oral drugs, and one [40] showed that LAN with a dose of 200 MU including the trigonum and LAN with a dose of 300 MU excluding the trigonum produce identical efficacy and safety. One randomised controlled trial [41] showed that LAN with a dose of 200 MU combined with electroacupuncture is more effective than LAN with a dose of 200 MU alone.

#### 2.2.8. Gastroenterological Indications

Gastroenterological indications are a collection of mainly experimental indications with achalasia being the most robust one studied. As shown in Table 1, altogether five LAN publications refer to gastroenterological indications. One publication as a primary topic covers gastroparesis, two publications are about achalasia (both as primary topic publications), one publication is about oesophageal strictures (as a primary topic publication) and one publication is about dysphagia (as a primary topic publication). One randomised controlled trial [42] indicated that LAN is effective and safe for the prevention of oesophageal strictures. All other studies are either case studies or observational studies.

#### 2.2.9. Other Indications

There are publications dealing with Raynaud syndrome (as a primary topic publication) [43], Anismus (as a primary topic publication) [44] and Tinnitus (as a primary topic publication) [45]. The studies are case studies or observational studies and showed LAN is effective and safe.

### 2.3. Glandular Indications

Glandular indications are indications where the therapeutic effect is based on an activity reduction of exocrine glands innervated by the cholinergic autonomic nervous system. Affected glands may be sweat glands, salivary glands and lacrimal glands. Additional exocrine glands may be used for therapeutic purposes in the future. Injections into smooth muscles which are also innervated by the cholinergic autonomic nervous system will be covered in the section “Muscle indications”. Glandular indications are very rewarding indications for BT therapy. Especially, hyperhidrosis is an important indication because of its prevalence, the lack of successful treatment alternatives and the excellent results of BT therapy. As shown in Table 1, altogether 16 publications refer to glandular indications.

#### 2.3.1. Hyperhidrosis

Eleven publications (all as primary topic publications) cover hyperhidrosis. Topics include general data on efficacy and safety, different LAN doses, different regions affected (axillary and palmar) and different aetiologies (primary and secondary). Often, unusual terminology is applied, such as bromhidrosis, osmidrosis and hircismus.

As shown in Table 6, two randomised controlled trials on axillary hyperhidrosisone of them [46] demonstrated that LAN with a dose of 200 MU per axilla has longer efficacy than LAN with a dose of 50 MU per axilla. The other randomised controlled trial [47] suggested that LAN is more effective for mild and moderate axillary hyperhidrosis, whereas surgical gland excision is more effective for severe one.

#### 2.3.2. Sialorrhea

Four studies (all as primary topic publications) deal with sialorrhea or hypersalivation. Topics cover general data on efficacy and safety, ultrasound guidance of parotid and submandibular gland injections and different aetiologies (cerebral palsy and brain damage). As shown in Table 6, one randomised controlled trial [48] showed that the electrical stimulation of tongue, orbicular oris and buccinators muscles with LAN is more effective than electrical stimulation alone.

#### 2.3.3. Prostate Hyperplasia

As shown in Table 6, one observational study [49] suggested the effect of LAN on benign prostate hyperplasia. The mechanism involved remains unclear.

### 2.4. Pain Indications

Pain indications are indications where the therapeutic effect is based upon modulations of nociception, pain transmission and pain processing. Involved neural structures are not entirely understood. Where and how BT might interfere is not well investigated. Central nervous system effects may exist. BT can also achieve pain reduction by reduction excessive muscle hyperactivity. Additional nociceptive effects may be caused by potentially anti-inflammatory effects.

Pain indications are the latest major extension of BT therapy with chronic migraine being all dominant. As shown in Table 1, altogether 33 LAN publications refer to pain indications. Twenty publications (all as primary topic publications) cover migraine, nine publications are about trigeminus neuralgia (all as primary topic publications), and three publications are about postherpetic neuralgia (all as primary topic publications).

#### 2.4.1. Migraine

Table 7 shows four randomised controlled trials dealing with migraine: one of them [50] showed the efficacy and safety of LAN against placebo, and one study [51] demonstrated that LAN is more efficient than lidocaine combined with prednisolone injections. one study [52], the authors found that LAN with oral drugs is more effective than oral drugs alone, and one publication [53] showed that LAN application in acupuncture points is more efficient than application-fixed injection sites. Three randomised controlled trials deal with chronic migraine: one of them [54] demonstrated the efficacy and safety of LAN against placebo, and the other two [55,56] showed that LAN combined with infrared light is more efficient than LAN alone, infrared light alone and oral drugs alone.

#### 2.4.2. Trigeminal Neuralgia

Three randomised controlled trials examined LAN in trigeminal neuralgia: one of them showed efficacy and safety against placebo [57], one [58] showed that LAN with a dose of 75 MU is not more effective than LAN with a dose of 25 MU, and one [59] found that doubling the LAN dose as a booster injection does not increase efficacy.

#### 2.4.3. Tension-Type Headache

One randomised controlled trial [60] found that LAN is more effective than oral drugs in tension-type headaches.

#### 2.4.4. Post-Herpetic Neuralgia

One randomised controlled trial [61] demonstrated the efficacy and safety of LAN against placebo in post-herpetic neuralgia.

### 2.5. Aesthetic Indications

Aesthetic indications are indications where BT is not used to treat a disease, but to improve appearance. All indications presented here are medical procedures reserved for physicians. They are generally not covered by reimbursement systems. Aesthetic indications were developed very early on by neurologists treating blepharospasm. Soon, their potential for aesthetic medicine was realised and strategically developed. Around 50% of global medical BT use is in the field of aesthetic medicine. The contouring of the masseter and the calf muscles is specific to Asian countries. Altogether, 48 LAN publications refer to aesthetic indications. Thirty-eight publications (37 as primary topic publications and one as a secondary topic publication) cover wrinkles, six publications refer to calf reduction (all as primary topic publications), two publications refer to masseter hypertrophy (both as primary topic publications), two publications are about scars (both as primary topic publications) and one publication is about acne (as a primary topic publication). As shown in Table 8, all five randomised controlled trials studied LAN in forehead wrinkles and glabellar lines. One study [62] showed LAN’s efficacy and safety against placebo in forehead wrinkles. Four of them [63,64,65,66] showed that lower LAN doses produce identical peak efficacy, but shorter effect duration in forehead wrinkles and glabellar lines at a slightly less risk of adverse effects.

### 2.6. Methods

As shown in Table 1, six publications refer to methodological issues. Three publications (as primary topics) cover LAN comparisons with ONA and ABO, one publication refers to remote effects (as a primary topic), and two publications are about allergic effects (as primary topics). All comparative studies suggest similar efficacy and safety, when LAN is converted to ONA at a 1:1 LAN–ONA dose conversion ratio [10] and to ABO at a 1:3 LAN–ABO dose conversion ratio [67]. Rare allergic reactions contribute to LAN’s bovine gelatine content [68,69].

### 2.7. Reviews and Guidelines

Altogether, 20 publications are reviews and guidelines. Eighteen reviews cover BT use in China in areas of neurology, cervical dystonia, autonomic disorders, hyperhidrosis, urology, overactive bladder, prostate hyperplasia, scars, aesthetic applications, calf hypertrophy and depression. Two guidelines deal with BT therapy in plastic surgery and adult limb spasticity.

## 3. Discussion

This paper presents the first complete English bibliography of all LAN publications until 1 July 2018. It is based on the English database PubMed and the Chinese data base Science and Technology Paper Citation Database (STPCD). It shows, for the first time, all authors’ names, publication titles and source names in English, thus making them accessible to the international English-speaking audience. Appendix A can be found.

This paper also presents the first comprehensive review of all LAN publications until 1 July 2018. It is based on the English bibliography above and additional English translations of all LAN articles originally published in Chinese. It is structured as a formalised literature review.

LAN publications have been produced over a period of more than 20 years, making LAN one of the three longest used BT drugs worldwide.

LAN is used for motor, glandular, pain and aesthetic indications. Medical specialties involved are neurology, aesthetics, paediatrics, ophthalmology, urology, gastroenterology, ENT and surgery. Therefore, LAN is used in all major BT indications other BT drugs have been used for. This also documents the widespread use of LAN in China.

Of the 379 publications retrieved, 71 are randomised controlled trials, 41 are interventional studies, 231 are observational studies, 15 are case studies, 18 are reviews, and 2 are guidelines. This constitutes a publication basis similar to the ones of other international BT drugs. Relatively low percentages of randomised controlled trials and interventional studies are also typical for other BT drugs.

None of the publications presents data contradictory to those generated for other BT type-A drugs. Based on a 1:1 LAN–ONA conversion ratio, LAN seems to have a similar efficacy and safety features compared to ONA. A recent publication studying the biological potency of LAN, ONA and INCO in a direct comparison did not detect differences in their potency labelling [70], thus confirming this equipotency.

Large controlled multicentre studies will eventually become necessary for LAN’s registrations in Europe and North America. Further studies on LAN’s product details, including its specific biological potency, its production consistency and its stability, will then become necessary. LAN’s increased international presence will increase the number of international studies published in international journals.

## 4. Methods

This review is based upon comprehensive literature searches in the English data base “PubMed” (National Center of Biomedical Information, United States National Library, Medicine and National Institutes of Health) and the Chinese data base “STPCD” within Wanfang data. The searches include all references published until 1 July 2018. They were performed along two axes with the following search words:

Axis 1: Botulinum toxin and China or Chinese or Lanzhou, Prosigne, Lantox, Hengli, HBTX-A, Lanzox, Redux, Liftox, CBTX-A.

Axis 2: Strabismus, ophthalmology, blepharospasm, hemifacial spasm, Meige syndrome, spasmodic dysphonia, torticollis, cervical dystonia, craniocervical dystonia spasticity, writer’s cramp, dystonias, hemifacial spasm, tics, spasmodic dysphonia, spasticity, cerebral palsy, strabismus, bladder dysfunction, temporomandibular joint disorder, gastroenterology, gastroparesis, achalasia, dysphagia, anismus, tinnitus, hyperhidrosis, sialorrhea, Raynaud syndrome, headache, trigeminus neuralgia, postherpetic neuralgia, urology, bladder dysfunction, aesthetic, masseter, wrinkles, gastrocnemius, masseter, scars, adverse effects, allergic reaction.

All publications contain at least one item on each axis. All retrieved publications appearing in Chinese were first translated into English, before they were referenced with their authors’ names, publication year, study design, title and source. Within their sections, all references were ordered by their authors’ names using the English alphabet and by their publication years. The complete bibliography is presented as Appendix A. Each reference was classified under the terms motor indications, glandular indications, pain indications, aesthetic indications, methods and reviews and guidelines, as shown in Table 1. All publications were further classified according to their study design as randomised controlled trial, interventional study, observational study and case study, review and guideline. In Table 1, all references were also classified into primary topic publications where the topic is the primary one and secondary topic publications where the topic is an additional one. Selected publications classified as randomised controlled trial, interventional study, review and guideline may be discussed in detail in the main text and the tables of this article. Discussions will include efficacy and adverse effects.

## Figures and Tables

**Table 1 toxins-13-00370-t001:** Overview over all lanbotulinumtoxinA (LAN) publications retrieved from Pubmed and Science and Technology Paper Citation Database.

Motor Indications									
Indication	S	PT	ST	All	RCT	IS	OS	CS	RG
Blepharospasm	N	32	3	35	2	6	27		
Meige syndrome	N	13	4	17		1	15	1	
Cervical dystonia	N	32	0	32	7	1	24		
Craniocervical dystonia	N	5	0	5		1	4		
Writer’s cramp	N	2	0	2			2		
Dystonias	N	11	1	12			12		
Spasmodic dysphonia	E	3	0	3	1		2		
TMJ disorder	N	3	0	3	1		2		
Hemifacial spasm	N	17	18	35	2	3	30		
Tics	N	1	0	1			1		
Spasticity	N	40	0	40	18	7	14	1	
Cerebral Palsy	P	38	0	38	16	7	15		
Strabismus	O	17	1	18	1	2	15		
Bladder dysfunction	U	8	0	8	3	1	4		
Gastroparesis	G	1	0	1				1	
Achalasia	G	2	0	2				2	
Oesophageal strictures	G	1	0	1	1				
Dysphagia	G	1	0	1				1	
Anismus	S	1	0	1			1		
Raynaud syndrome	N	1	0	1				1	
Tinnitus	E	1	0	1				1	
**Glandular indications**									
Hyperhidrosis	N	11	0	11	2	2	7		
Sialorhea	N	4	0	4	1		3		
Prostate hyperplasia	U	1	0	1				1	
**Pain indications**									
Migraine	N	20	0	20	7	2	11		
Trigeminal neuralgia	N	9	0	9	3	2	3	1	
Postherpetic neuralgia	N	3	0	3	1		1	1	
**Aesthetic indications**									
Wrinkles	A	36	1	37	5	2	30		
Calf reduction	A	6	0	6		1	5		
Masseter reduction	A	2	0	2			2		
Scars	A	2	0	2			1	1	
Acne	A	1	0	1				1	
**Methods**									
Drug comparison		3	0	3		2	1		
Remote effects		1	0	1		1			
Allergic reactions		2	0	2				2	
**Reviews and guidelines**									
Reviews									18
Guidelines									2
All		351	28	379	71	41	232	15	20

S, medical specialty (A, aesthetics; E, ear, nose and throat; G, gastroenterology; N, neurology; O, ophthalmology; P, paediatrics; S, surgery; U, urology; PT, primary topic publication; ST, secondary topic publication; All, all publications; RCT, randomised controlled trial; IS, interventional study; OS, observational study; CS, case study; R, review; G, guideline; TMJ, temporomandibular joint.

**Table 2 toxins-13-00370-t002:** LAN publications on dystonia and hemifacial spasm.

Authors	I	D	n	Methods	Results
Quagliato et al. [10]	BS HFS	RCT	57	Comparing LAN and ONA. BS = 21, HFS = 36. LAN = 29, ONA = 28. LAN dose: ONA dose ratio = 1:1. Total BS dose: 60 MU. Total HFS dose: 35 MU	No differences in efficacy and AE
Lu et al. [11]	BS	RCT	85	Comparing traditional injection sites and injection sites sparing medial lower eyelids	Sparing medial lower eyelids improves tear film stability and lacrimal fluid drainage
Rieder et al. [12]	BS HFS	IS	26	Comparing LAN and ONA. Cross-over design. BS = 8, HFS = 18. LAN dose:ONA dose ratio = 1:1. Total dose: 2.5–5 MU per injection site	No differences in efficacy and AE
Peng et al. [20]	HFS	RCT	63	Comparing effects of traditional injection sites and those of traditional injection sites with an additional dose of 4 MU in posterior auricular muscle (PAM). HFS with auricular symptoms	Additional PAM injections reduce auricular symptoms
Li et al. [21]	HFS	IS	20	Comparing LAN with a dose of 50 MU/mL and LAN with a dose of 25 MU/mL. Cross-over design. LAN dose: 2.5–5 MU in each injection site. Washout period: 12 months	No difference in efficacy. Longer duration and more AE with higher LAN concentration
Barbosa et al. [3]	CD	RCT	34	Comparing LAN and ABO. ABO = 14. LAN = 20. ABO dose:LAN dose ratio = 3:1. Follow-up for 5 injection series every 3 months.	No difference in efficacy, effect duration and AE
Quagliato et al. [13]	CD	RCT	24	Comparing LAN and ONA. LAN dose–ONA dose ratio = 1:1. Total dose: 300 MU	No difference in efficacy, effect duration and AE
Huang et al. [14]	CD	RCT	105	Comparing oral drugs, LAN and LAN with orthopaedic joint brace (external fixator for head, neck, chest and back). Ultrasound guidance	LAN with orthopaedic joint brace is better than LAN alone. Oral medication without efficacy
Wu et al. [15]	CD	RCT	68	Comparing LAN with EMG guidance and LAN without EMG guidance.	EMG guidance with prolonged efficacy and less AE, but more injection site pain. Maximal efficacies are not different.
Hu et al. [16]	CD	RCT	126	Comparing LAN with a dose of25 MU/mL and LAN with a dose of 17 MU/mL	No difference in efficacy and AE
Luo et al. [17]	CD	RCT	27	Comparing LAN with a dose of 50 MU/mL and LAN with a dose of 12.5 MU/mL	No difference in efficacy and AE
Hu et al. [18]	SD-AD	RCT	27	Comparing LAN with a dose of 5 MU and LAN with a dose of 10 MU. Blitzer grade: ≥III. Blitzer dose: 25 MU/mL. EMG guidance. Unilateral thyroarytenoid injections	No difference in onset time, time-to-peak effect and AE. Higher dose produces a longer effect duration
Jiang & You [19]	TMJD	RCT	90	Comparing semiconductor laser treatment and semiconductor laser treatment with LAN injection	Semiconductor laser therapy with LAN has a better efficacy on inflammation and pain than semiconductor laser treatment alone.

I, indication; D, design; n, number of patients/controls; RCT, randomised controlled trial; BS, blepharospasm; HFS, hemifacial spasm; CD, cervical dystonia; SD-AD, spasmodic dysphonia (adductor type); IS, interventional study; AE, adverse effects; LAN, Lantox^®^; TMJD, temporomandibular joint dysoorder.

**Table 3 toxins-13-00370-t003:** LAN publications on spasticity.

Authors	I	D	n	Methods	Results
Yang et al. [22]	AS	RCT	178	Comparing LAN and placebo. Multi-centre trial. LAN = 118. placebo = 60. EMG guidance. LAN dose: 200 MU. An additional dose of 40 MU for thumb spasticity if necessary	LAN is effective and safe. No difference in AE
Cui & Zhang [23]	AS	RCT	43	Comparing LAN with rehabilitation and rehabilitation alone	LAN with rehabilitation reduces MAS and ROM more effectively
Lan et al. [24]	AS	RCT	32	Comparing LAN with rehabilitation and rehabilitation alone	LAN with rehabilitation is more effective.
Wang [25]	AS	RCT	60	Comparing LAN with rehabilitation and rehabilitation alone	LAN with rehabilitation is more effective.
Zhang [27]	AS	RCT	60	Comparing LAN with acupuncture and acupuncture alone	LAN with acupuncture is more effective.
Lai et al. [26]	LS	RCT	109	Comparing rehabilitation with LAN and rehabilitation without LAN	Rehabilitation with LAN works better.
Ding et al. [28]	LS	RCT	103	Comparing LAN with rehabilitation and LAN with ankle foot brace and LAN without ankle foot brace	LAN with ankle foot brace and rehabilitation works best.
Ding et al. [29]	LS	RCT	80	ComparingLAN with rehabilitation with spasmodic muscle therapy and LAN with rehabilitation without spasmodic muscle therapy	LAN with rehabilitation and with spasmodic muscle therapy works better and longer.
Li et al. [30]	LS	RCT	40	Comparing LAN with rehabilitation and LAN with intensive rehabilitation	LAN with intensive rehabilitation works better.
Li et al. [31]	LS	RCT	104	Comparing different concentrations (50 MU/mL or 100 MU/mL) and doses (200 MU or 400 MU) of LAN	High-dose/low-concentration combination produces the best effect.

I, indication; D, design; n, number of patients/subjects; RCT, randomised controlled trail; AE, adverse effects; LAN, Lantox^®^; AS, arm spasticity; LS, leg spasticity.

**Table 4 toxins-13-00370-t004:** LAN publications on cerebral palsy.

Authors	I	D	n	Methods	Results
Xu et al. [32]	CP	RCT	43	Comparing LAN with rehabilitation and rehabilitation alone. Children. EMG guidance	LAN with rehabilitation works better.
Duan & Zhang [33]	CP	RCT	80	Comparing rehabilitation with LAN and rehabilitation alone	LAN with rehabilitation has a better efficacy.
Liu et al. [34]	CP	RCT	244	Comparing LAN with rehabilitation and LAN without rehabilitation. Patient age: 1–23 years	LAN with rehabilitation works better.
Niu et al. [35]	CP	RCT	256	Comparing a LAN dose of 3 MU/kg and a LAN dose of 6 MU/kg	In muscle tension levels III-IV, the high-dose group has a better efficacy.
Peng & Cai [36]	CP	RCT	90	Comparing a LAN dose of 3 MU/kg and LAN doses of 4 MU/kg and 5 MU/kg. EMG guidance	No difference
Liang et al. [37]	CP	RCT	64	Comparing LAN with TCM with Ueda * physiotherapy and LAN with TCM with Bobath physiotherapy	LAN with Ueda physiotherapy and TCM improves CP more effectively.

I, indication; D, design; n: number of patients/subjects; RCT, randomised controlled trail; AE, adverse effects; LAN, Lantox^®^; CP, cerebral palsy. TCM, traditional Chinese medicine. *, traditional Japanese Kampo drug against spasticity.

**Table 5 toxins-13-00370-t005:** LAN publications on bladder dysfunctions.

Authors	I	D	n	Methods	Results
Li et al. [39]	OB	RCT	24	Comparing LAN with a dose of 100 MU and oral drugs. Only females. Target muscle: detrusor excluding trigonum	LAN is more effective.
Fu et al. [40]	NI	RCT	60	Comparing LAN with a dose of 200 MU including trigone and LAN with a dose of 300 MU excluding trigonum. Follow-up at 4 weeks	No difference in efficacy and AE
Meng et al. [41]	NB	RCT	35	Comparing a LAN dose of 200 MU with electro-acupuncture and LAN alone. Transperineal application into external urethral sphincter	Combined therapy is more effective.

I, indication; D, design; n, number of patients/controls; RCT, randomised controlled trial; AE, adverse effects; LAN, Lantox^®^; OB, overactive bladder; NI, neurogenic Incontinence; NB, neurogenic bladder.

**Table 6 toxins-13-00370-t006:** LAN publications on glandular indications.

Author	I	D	n	Methods	Results
Gao et al. [46]	AH	RCT	92	Comparing LAN with a dose of 200 MU and LAN with a dose of 50 MU	A higher LAN dose has a longer efficacy.
Xie et al. [47]	BH	RCT	150	Comparing LAN and surgical gland excision	LAN is more effective for mild and moderate BH and less effective for severe BH.
Peng et al. [48]	S	RCT	98	Comparing LAN with electrical stimulation and electrical stimulation alone	Combination therapy is more effective than electrical stimulation.
Ding et al. [49]	BPH	OS	32	LAN with a dose of 200 MU was injected into five points at the lateral and middle lobes of the prostate under the guidance of ultrasound with a balloon dilatational device.	All symptoms and indicators are improved and maintained for a period of at least 1 year.

I, indication; D, design; n, number of patients/subjects; RCT, randomised controlled trials; AE, adverse effects; LAN, Lantox^®^; AH, axillar hyperhidrosis; S, sialorhea; BH, bromhidrosis; BPH, benign prostatic hyperplasia.

**Table 7 toxins-13-00370-t007:** LAN publications on pain indications.

Authors	I	D	n	Methods	Results
Ge et al. [50]	M	RCT	79	Comparing LAN and placebo	LAN is more efficient. No difference in AE
Li et al. [51]	M	RCT	61	Comparing LAN and lidocaine with prednisolone injections	LAN is safe and more efficient.
Lu & Xu [52]	M	RCT	94	Comparing LAN with oral drugs and oral drugs alone	LAN is more efficient.
Shao et al. [53]	M	RCT	60	Comparing LAN in fixed injection sites and LAN in acupuncture points. LAN dose: 25 MU	LAN in acupuncture points is more effective.
Wang et al. [54]	CM	RCT	101	Comparing LAN and placebo. Injection sites fixed (frontalis, temporalis and occipitalis) plus pain sites	LAN is more efficient. No relevant AE
Song et al. [55]	CM	RCT	91	Comparing LAN with infrared light and LAN alone and infrared light alone and oral drugs alone	Combined therapy is more efficient than oral medication alone, infrared light alone and LAN alone.
Gu & Feng [56]	CM	RCT	56	Comparing LAN with infrared light and LAN alone	LAN with infrared light is more effective.
Wu et al. [57]	TN	RCT	42	Comparing LAN and placebo	LAN is safe and more effective.
Zhang et al. [58]	TN	RCT	84	Comparing LAN with a dose of 25 MU and LAN with a dose of 75 MU and placebo	LAN is safe and effective. No difference between dose groups
Zhang et al. [59]	TN	RCT	100	Comparing LAN and LAN with booster injections. Booster dose identical to initial dose	Booster injections are without advantage. No difference in AE
Zhai et al. [60]	TTH	RCT	50	Comparing LAN and oral drugs	LAN is more effective. Temporary neck weakness
Xiao et al. [61]	PHN	RCT	60	Comparing LAN and lidocaine and placebo	LAN reduces pain and increases sleeping time

I, indication; D, design; n, number of patients/subjects; RCT, randomised controlled trial; AE, adverse effects; LAN, Lantox^®^.

**Table 8 toxins-13-00370-t008:** LAN publications on aesthetic indications.

Author	I	D	n	Methods	Results
Zhu et al. [62]	FW	RCT	40	Comparing LAN and placebo	LAN is effective and safe in FW.
Li [63]	FW	RCT	132	Comparing LAN with a dose of 50 MU/mL and LAN with a dose of 25 MU/mL. Identical volume	The Effect duration is longer and AE are more with a LAN dose of 50 MU/mL. No difference in efficacy
Zhu et al. [64]	FW	RCT	107	Comparing a LAN dose of 50 MU/mL and a LAN dose of 25 MU/mL. Identical volume	LAN with a dose of 50 MU/mL has longer effect duration and more AE. No difference in peak efficacy
Feng et al. [65]	GL	RCT	488	Comparing LAN with a dose of 10 MU and LAN with a dose of 20 MU and placebo	LAN with a dose of 20 MU has a stronger effect and more AE.
Wang et al. [66]	FW	RCT	86	Comparing LAN doses of 20–35 MU and LAN doses of 45–60 MU	Efficacy is identical. LAN with doses of 20–35 MU produces less AE.

I, indication; D, design; n, number of patients/subjects; RCT, randomised controlled trial; LAN, Lantox^®^; AE, adverse effects; FW, facial wrinkles; GL, glabella lines.

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
