# Peer review of "Lantox—The Chinese Botulinum Toxin Drug—Complete English Bibliography and Comprehensive Formalised Literature Review"

_toxins, 2021, doi:10.3390/toxins13060370_

Round 1

Reviewer 1 Report

This review presents the first complete English bibliography of all publications up to July 1st 2018 concerning Lantox (LAN) - the Chinese Botulinum Toxin Drug. This review is based on the English data base PubMed and the Chinese data base Science and Technology Paper Citation Database. It shows, for the first time, all authors' names, publication titles and source names in English and, thus, making them accessible to the English speaking international audience. The whole list can be found as supplementary material to this article.

This review is interesting and new. Nevertheless, this manuscript needs substantial improvement and correction before publishing may be possible.

General points:

Please add list of abbreviations to your manuscript.

Please include in your review all publications until the 2021, not only up to July 1st 2018. The review should be as informative as possible, i.e., must be up to date.

Special points:

Introduction

The Introduction section should be substantially improved, i.e. by substantial references in the field:

Line 28: please add references at the end of this sentence.

Line 29: please add references at the end of this sentence.

Lines 31-32: please add multiple references at the end of this sentence.

Lines 32-36: please add more references at the end of this sentence.

Lines 37-39: please add references at the end of this sentence.

Lines 39-42: please add references at the end of this sentence.

Lines 42-45: please add references at the end of this sentence.

Line 46: please add references at the end of this sentence.

Lines 47-50: please add references at the end of this sentence.

Lines 56-58: please add references at the end of this sentence.

Lines 88-89: please add references at the end of this sentence.

Lines 90-93: please add references at the end of each these sentences.

Line 104: please check your font size.

Lines 110-114: please add the examples of corresponding references.

Lines 115-127: you said: Thirty-two publications (all as primary topic) are covering cervical dystonia. They are dealing with injection guidance by electromyography or ultrasound, variations of LAN concentrations, drug comparisons between LAN, ONA and ABO, a combination of LAN and orthoses, a combination of LAN and thermocoagulation of peripheral nerves, treatment of tardive cervical dystonia, as well as long-term studies and reports on larger cohorts. Table 2 shows 6 selected RCT for cervical dystonia.

Please create for the remaining 26 publications about cervical dystonia the same table as you did for 6 selected publications in the main body of the ms, and add this table to your Supplement to the existing reference list.

Lines 140-148: please create the same table for hemifacial spasm as you did for cervical dystonia und add this table to the main part of your review.

Table 2: please add to your “n” column the exactly number of female and male patients for each single publication included.

Line 158: please add references at the end of this sentence.

Lines 158-159: please add references at the end of this sentence.

Lines 160-165: please add references at the end of this sentence.

Table 3: please add to your “n” column the exactly number of female and male patients for each single publication included.

Lines 178-179: please add references at the end of this sentence.

Lines 179-180: please add references at the end of this sentence.

Lines 181-183: please add references at the end of this sentence.

Table 4: please add to your “n” column the exactly number of female and male patients for each single publication included.

Please create for the remaining 32 publications about cerebral palsy the same table as you did for 6 selected publications in the main body of the ms, and add this table to your Supplement to the existing reference list.

Lines 199-200: please add references at the end of this sentence.

Line 208: please add references at the end of this sentence.

Lines 199-206: please create the same table for strabismus as you did for cervical dystonia und add this table to the main part of your review.

Table 5: please add to your “n” column the exactly number of female and male patients for each single publication included.

Lines 221-231: please create the same table for gastroenterological indications as you did for cervical dystonia und add this table to the main part of your review.

Lines 233-235: please add references at the end of this sentence.

Lines 235-239: please add references at the end of each these sentences.

Table 6: please add to your “n” column the exactly number of female and male patients for each single publication included.

Lines 255-259: please create the same table for sialorrhea as you did for cervical dystonia und add this table to the main part of your review.

Lines 260-262: please create the same table for prostate hyperplasia as you did for cervical dystonia und add this table to the main part of your review.

Please create for the remaining 11 publications about pain the same table as you did for 12 selected publications in the main body of the ms, and add this table to your Supplement to the existing reference list.

Lines 264-269: please add references at the end of each these sentences.

Lines 278-285: please create the same table migrane as you did for cervical dystonia und add this table to the main part of your review.

Lines 287-290: please create the same table for trigeminal neuralgia as you did for cervical dystonia und add this table to the main part of your review.

Lines 298-300: please add references at the end of each these sentences.

Please create for the remaining 43 publications about aesthetic indications the same table as you did for 5 selected publications in the main body of the ms, and add this table to your Supplement to the existing reference list.

Discussion

Please add to your Discussion part the comparison of action and effects between the Lantox (LAN) and common used three major BT drugs available: onabotulinumtoxinA (ONA, Bo-33 tox®, Allergan Inc, Irvine, CA, USA), abobotulinumtoxinA (ABO, Dysport®, Beaufour Ip-34 sen, Boulogne-Bilancourt, France) and incobotulinumtoxinA (INCO, Xeomin®, Merz 35 Pharmaceuticals GmbH, Frankfurt/M, Germany).

Methods

Lines 364-365: you said: The searches include all references published up to July 1st 2018.

Please include in your review all publications until the 2021, not only up to July 1st 2018.

Reviewer 2 Report

The authors present a useful overview of published work on LANTOX, a botulinum product. The review is well written but some corrections are required.

  1. ONA, Bo-33 tox®, Allergan Inc, Irvine, CA, USA - correct to Abbvie
  2. p.2 When describing LAN, please mention excipients such as LAN's bovine gelatine content.
  3. p. 3 ‘231 publications were OS, 71 RCT, 41 IS, 18 R, 15 CS and 2 G.’ This needs to be spelled her. Generally, one common table of used abbreviations is needed (e.g. Migraine – M, etc.)
  4. A separate section on reported Adverse Effects is missing
  5. Shujin Huolo lotion – is not clear
  6. p. 7 ‘1 RCT [28] suggested that using higher LAN doses and lower concentrations may improve LAN therapy outcome’ - higher LAN doses and lower concentrations – not clear
  7. traditional Chinese medicine and Ueda could improve – what is Ueda?
  8. Some references are missing in the text. For example on p.9: ‘One publication each is dealing with Raynaud syndrome (as primary topic), Anismus (as primary topic) and Tinnitus (as primary topic). The studies are CS or OS and showed LAN is effective and safe.’

Author Response

Reviewer#2

The authors present a useful overview of published work on LANTOX, a botulinum product. The review is well written but some corrections are required.

Issue #1: ONA, Bo-33 tox®, Allergan Inc, Irvine, CA, USA - correct to Abbvie

Reply: Point well taken as the company name changed after submission. The requested change was done.

Issue #2: p.2 When describing LAN, please mention excipients such as LAN's bovine gelatine content.

Reply: Point well taken. The Introduction now contains the following passage on the comparison of LAN with other BT drugs including its gelatine content and potency labelling: These basic components are similar to ONA and ABO. However, the manufacturing process differs and the stabilising protein used in LAN is not human serum albumin, but bovine gelatine 5mg per vial. Other excipients include dextran 25mg and sucrose 25mg per vial. The role of gelatine as well as LAN's potency labelling will be discussed below. BT's clinical pharmacology was recently reviewed in detail (Dressler 2021).

Issue #3: p. 3 ‘231 publications were OS, 71 RCT, 41 IS, 18 R, 15 CS and 2 G.’ This needs to be spelled her. Generally, one common table of used abbreviations is needed (e.g. Migraine – M, etc.)

A separate section on reported Adverse Effects is missing

Reply: As suggested, we reduced the number of abbreviations. OS, RCT, IS, CS G are dropped. On the request of another reviewer an abbreviation table was also introduced in the Methods section. Adverse effects are discussed together with efficacy for all selected publications discussed in detail. The following passage was added to the ms: Selected publications classified as Randomised Controlled Trials, Interventional Studies or Reviews and Guidelines may be discussed in detail in the main text and the tables of this article. Discussions will include efficacy and adverse effects.

Issue # 4: Shujin Huolo lotion – is not clear

Reply: We clarified with the following passage: Shujin Huolo lotion (a traditional Chinese medicine to reduce spasticity)

Issue #5: p. 7 ‘1 RCT [28] suggested that using higher LAN doses and lower concentrations may improve LAN therapy outcome’ - higher LAN doses and lower concentrations – not clear

Reply: This was a four-armed study with high dose/low concentration, high dose/high concentration, low dose/low concentration and low dose/high concentration.

Issue #6: traditional Chinese medicine and Ueda could improve – what is Ueda?

Reply: We clarified with the following passage: *a combination of the Japanese antispastic Kampo drug Ueda and physiotherapy.

Issue #7: Some references are missing in the text. For example on p.9: ‘One publication each is dealing with Raynaud syndrome (as primary topic), Anismus (as primary topic) and Tinnitus (as primary topic). The studies are CS or OS and showed LAN is effective and safe.’

Reply: We added the requested refs.

Reviewer 3 Report

As far as the reviewer knows, the main excipients of LAN (per 100 U vial) are gelatine 5 mg, dextran 25 mg, and sucrose 25 mg with a minimum protein load of 4–5 ng/100 units. While most other botulinum toxin preparations use human serum albumin to prevent the neurotoxin from adhering to the wall of the vial or syringe, LAN uses bovine gelatin which may be more prone to trigger an immunological response and allergic reactions (as described in lines 320-321). Allergic reactions, as reported in the literature, are  possible but rare after botulinum neurotoxin injection; however, the reviewer recommends the authors include one or two sentences providing clear information on the excipients of LAN (preferably at the end of line 61).

Round 2

Reviewer 1 Report

I very carefully check the revised version with the marked (yellow) inclusions.

Although some aspects of my first review were included, many are still not. Thus, this is the review of the revised version of the manuscript of your mail.

Review of 1172173-R1 Unfortunately, the authors were still not willing to correct and improve this manuscript according to all my previous proposals. Therefore, this manuscript still needs substantial improvements and corrections before publishing may be possible.

Please note that I refer to the line numbers given at the right page side of the manuscript, where they are clearly indicated.

General points: Please include in your review all publications until the 2021, not only up to July 1st 2018. The review should be as informative as possible, i.e., must be up to date.

The spaces between the words and numbers and spell check required in the whole manuscript.

Special points:

Key Contributions: please add the Key Contributions to your manuscript.

Introduction

The Introduction section should be substantially improved, i.e. by substantial references in the field:

All lines numbers please see in your last manuscript version in the right side of each page.

Line 42: please add references at the end of this sentence.

Line 43: please add references at the end of this sentence.

Lines 45-46: please add multiple references at the end of this sentence.

Lines 47-50: please add more references at the end of this sentence.

Lines 50-53: please add references at the end of this sentence.

Lines 54-55: please add references at the end of this sentence.

Lines 55-56: please add references at the end of this sentence.

Lines 56-57: please add references at the end of this sentence.

Lines 59-62: please add references at the end of this sentence.

Lines 62-63: please add references at the end of this sentence.

Lines 65-67: please add references at the end of this sentence.

Lines 73-74: please add more references at the end of this sentence.

Lines 74-75: please add references at the end of this sentence.

Lines 78-79: please add references at the end of this sentence.

Lines 79-80: please add references at the end of this sentence.

Lines 80-81: please add references at the end of this sentence.

Results

Lines 112-113: please add references at the end of this sentence.

Lines 113-114: please add references at the end of this sentence.

Lines 114-115: please add references at the end of this sentence.

Lines 119-120: please add references at the end of this sentence.

Lines 120-121: please add references at the end of this sentence.

Lines 125-126: please add the examples of corresponding references.

Lines 136-137: please add references at the end of this sentence.

Line 139: please add the examples of corresponding references.

Lines 152-154: please add the examples of corresponding references.

Lines 159-160: please add these references described by you.

Lines 160-161: please add references at the end of this sentence.

Lines 166-167: please add references at the end of this sentence.

Lines 167-168: please add the examples of corresponding references.

Lines 181-182: please add references at the end of this sentence.

Lines 182-183: please add these references described by you.

Line 185: please add references at the end of this sentence.

Lines 186-193: please add the examples of corresponding references.

Lines 208-209: please add references at the end of this sentence.

Lines 209-211: please add references at the end of this sentence.

Lines 211-213: please add references at the end of this sentence.

Lines 213-214: please add the examples of corresponding references.

Lines Line 231: please add references at the end of this sentence.

Lines 232-233: please add the examples of corresponding references.

Line 241: please add references at the end of this sentence.

Line 242: please add references at the end of this sentence.

Line 246: please add these references described by you.

Lines 257-258: please add references at the end of this sentence.

Lines 258-260: please add corresponding references for each topic described by you in this sentence.

Line 262: please add the examples of corresponding references.

Line 276: please add the examples of corresponding references.

Line 291: please add the examples of corresponding references.

Lines 301-306: please add references at the end of these each sentences.

Lines 307-308: please add references at the end of this sentence.

Lines 336-342: please add references at the end of these each sentences.

Lines 362-366: please add the examples of corresponding references.

Discussion

Line 380: please add references at the end of this sentence.

Lines 380-382: please add references at the end of this sentence.

Lines 382-383: please add references at the end of this sentence.

Please add to your Discussion part the comparison of action and effects between the Lantox (LAN) and commonly used three major BT drugs available: onabotulinumtoxinA (ONA, Bo-33 tox®, Allergan Inc, Irvine, CA, USA), abobotulinumtoxinA (ABO, Dysport®, Beaufour Ip-34 sen, Boulogne-Bilancourt, France) and incobotulinumtoxinA (INCO, Xeomin®, Merz 35 Pharmaceuticals GmbH, Frankfurt/M, Germany).

Methods

Lines 403-404: you said: The searches include all references published up to July 1st 2018.

Please include in your review all publications until 2021, not only up to July 1st 2018.

Author Response

This manuscript is a resubmission of an earlier submission. The following is a list of the peer review reports and author responses from that submission.

Round 1

Reviewer 1 Report

This paper summarizes the literature on the Chinese botulinum toxin preparation, Lantox, including publications in Chinese that have generally been inaccessible to non-Chinese readers. It is important, to familiarize clinicians using botulinum toxin with Lantox.  However, the article itself is inconsistent in the presentation and it is not clear if the intent is to provide those outside of China with information on the nature of publications on Lantox, or if the intent is to summarize the results of studies in the Lantox literature. The paper needs to focus on one of these. As written, the paper intermixes these aspects and is confusing.

If the intent is to summarize the outcomes of studies in the Lantox literature and reach conclusions on Lantox efficacy/safety in relation to other available botulinum toxins, a structure/evidence-based review would be needed, which this paper is not. It includes all publications on Lantox for any indication.  The paper classifies publications by study design and presents select publications in tables with information on the indication, number of subjects, methods (purpose) and results.  However, there is no stated rationale for focusing on the particular trials presented in the tables. For some indications, like tics and GI indications, the article simply states the number of publications without information on the structure of the studies or outcomes.  For other indications, table 1 indicates more studies (especially RCTs) than are shown in the indication-specific table. For example, table 1 shows 18 RCTs for spasticity while table 3 lists only 10 of those RCTs.  Importantly, there appears to be no critical evaluation of the quality or validity of cited studies, as there would be if there were evidence tables similar to those used in systematic reviews.  Without that information, it is not possible to know if any of the study results are valid and reliable and there is inadequate support for the conclusion in the discussion that “Overall, LAN seems to have identical efficacy and safety features when compared to ONA based on a 1:1 conversion ratio.”

It the intent is to make those outside of China aware of publications on Lantox, the paper could be made much clearer and should focus on the nature and number of the publications and how to access them.   It would also be helpful to note which publications were head-to-head comparisons with other toxin formulations and whether Lantox appears to offer any advantage or has disadvantages compared to other formulations. 

Some minor points:

  • Not all abbreviations in the paper and tables are defined.
  • It is not clear what is meant by “interventional studies” as distinguished from RCTs, as RCTs are “interventional studies.”
  • The introduction is incorrect in saying that there are “3 major BT drugs available.” While there are 3 major type A botulinum toxins marketed, there is a 4th major drug: rimabotulinumtoxin B (Myobloc). 
  • In the introduction, it is interesting to say that Scott “invented” its BT therapeutic use…”. However, Kerner (who discovered BT as the cause of botulism) actually proposed in the 19th century that there was possible therapeutic use for BT in disorders of excessive muscle contraction.
  • The paper says that dystonia was the 2nd group of indications developed, but elsewhere says that hemifacial spasm was the 2nd indication developed. Those statements are inconsistent.

Author Response

Many thanks for your advice on our manuscript.

Reviewer 1

This paper summarizes the literature on the Chinese botulinum toxin preparation, Lantox, including publications in Chinese that have generally been inaccessible to non-Chinese readers. It is important, to familiarize clinicians using botulinum toxin with Lantox.

Issue #1: However, the article itself is inconsistent in the presentation and it is not clear if the intent is to provide those outside of China with information on the nature of publications on Lantox, or if the intent is to summarize the results of studies in the Lantox literature. The paper needs to focus on one of these. As written, the paper intermixes these aspects and is confusing.

Reply: The ms wants to allow access to the Chinese literature on Lantox. This is provided by the bibliography. It also provides a review of the Lantox-publications according to their types and their indications. This is provided by the main article. We, together with the other reviewers, don't believe this is confusing.

Issue #2: If the intent is to summarize the outcomes of studies in the Lantox literature and reach conclusions on Lantox efficacy/safety in relation to other available botulinum toxins, a structure/evidence-based review would be needed, which this paper is not.

Reply: As explained under Issue #1, it was the intention to summarize the Chinese Lantox-studies. It was not intended to compare different botulinum toxin preparations. BTW: what is a 'structure/evidence-based review'?

Issue #3: It includes all publications on Lantox for any indication.  The paper classifies publications by study design and presents select publications in tables with information on the indication, number of subjects, methods (purpose) and results.  However, there is no stated rationale for focusing on the particular trials presented in the tables.

Reply: As stated, tables show some selected RCT of the particular indications. Two IS and 1 OS were also included, as they were of particular interest. They may be deleted. Chief Editor, pls advise.

Issue # 4: For some indications, like tics and GI indications, the article simply states the number of publications without information on the structure of the studies or outcomes. 

Reply: As suggested, the additional information is now included in the ms as follows:

Tics: Tics have been tried successfully on several occasions, because of their similarity with dystonia, especially cranial dystonia. One LAN publication (as primary topic) refers to tics and is an OS.

Gastroenterological indications: Gastroenterological indications are a collection of mainly experimental indications with achalasia being the most robust one studied. Altogether 5 LAN publications refer to gastroenterological indications. One publication (as primary topic) is covering gastroparesis, 2 achalasia (both as primary topic), 1 oesophageal strictures (primary topic) and 1 dysphagia (as primary topic). One RCT (Wen et al. 2016) indicated that LAN is effective and safe for prevention of oesophageal strictures. All other studies are either CS or OS.

Issue # 5: For other indications, table 1 indicates more studies (especially RCTs) than are shown in the indication-specific table. For example, table 1 shows 18 RCTs for spasticity while table 3 lists only 10 of those RCTs.

Reply: To clarify this, the following passage is added in the Methods section: Selected publications classified as RCT, IS or R and G may be discussed in detail in the main text and the tables of this article.

Issue # 6: Importantly, there appears to be no critical evaluation of the quality or validity of cited studies, as there would be if there were evidence tables similar to those used in systematic reviews.  Without that information, it is not possible to know if any of the study results are valid and reliable and there is inadequate support for the conclusion in the discussion that “Overall, LAN seems to have identical efficacy and safety features when compared to ONA based on a 1:1 conversion ratio.”

Reply: For good reasons formalised reviews do not include personal opinions about the quality or soundness of the studies analysed.

Issue # 7: It the intent is to make those outside of China aware of publications on Lantox, the paper could be made much clearer and should focus on the nature and number of the publications and how to access them.   It would also be helpful to note which publications were head-to-head comparisons with other toxin formulations and whether Lantox appears to offer any advantage or has disadvantages compared to other formulations.

Reply: The nature and number of the publications are clearly described in the review part. Their access is shown in the bibliography part by the source attributed to each publication. Head-to-head comparisons are described in the tables, if they were based on RCT's.

Some minor points:

Issue #8: Not all abbreviations in the paper and tables are defined.

Reply: We only found TCM which stands for Traditional Chinese Medicine. This was corrected as suggested. Pls advise on others.

Issue #9: It is not clear what is meant by “interventional studies” as distinguished from RCTs, as RCTs are “interventional studies.”

Reply: RCT are interventional studies with a randomised, controlled design. IS are interventional studies without a randomised, controlled design.

Issue #10: The introduction is incorrect in saying that there are “3 major BT drugs available.” While there are 3 major type A botulinum toxins marketed, there is a 4th major drug: rimabotulinumtoxin B (Myobloc).

Reply: RimabotulinumtoxinB is clearly not a major BT drug. It is used for limited indications and in select situations only. It is associated with substantial adverse effects and has a highly relevant antigenicity problem. In many countries where it was originally registered it is no longer available. Its market share is negligible.

Issue #11: In the introduction, it is interesting to say that Scott “invented” its BT therapeutic use…”. However, Kerner (who discovered BT as the cause of botulism) actually proposed in the 19th century that there was possible therapeutic use for BT in disorders of excessive muscle contraction.

Reply: Kerner's statement was a pure and unsubstantiated speculation. Scott introduced BT's therapeutic use. He was, actually, not aware of Kerner's speculations.

Issue #12: The paper says that dystonia was the 2nd group of indications developed, but elsewhere says that hemifacial spasm was the 2nd indication developed. Those statements are inconsistent.

Reply: Both indications were first published in the same paper.

Reviewer 2 Report

This is a well written review on studies on the use of Botulinumtoxin in the Chinese literature. The review covers all different areas of botulinum treatment.

I have only one minor comment.

In tables on spasticity, pain and cosmetic use, the areas( leg, arm etc.) that are treated or the treatment indications (migraine, headache, maseter hypertrophy etc.) may be added, to make the table somewhat more informative.

Author Response

Many thanks for your advice on our manuscript.

Reviewer 2

This is a well written review on studies on the use of Botulinumtoxin in the Chinese literature. The review covers all different areas of botulinum treatment.

I have only one minor comment.

Issue #1: In tables on spasticity, pain and cosmetic use, the areas (leg, arm etc.) that are treated or the treatment indications (migraine, headache, maseter hypertrophy etc.) may be added, to make the table somewhat more informative.

Reply: In Table 3 (spasticity) the areas are given as AS/arm spasticity and LS/leg spasticity. In Table 7 (pain) the indications are given as M/migraine, CM/chronic migraine, TN/trigeminal neuralgia, TTH/tension type headache and PHN/postherpetic neuralgia. In Table 8 (aesthetic indications) the indications are given as FW/facial wrinkles and GL/glabella lines.

Reviewer 3 Report

The manuscript entitled “Lantox – The Chinese Botulinum Toxin Drug. Complete English Bibliography and Formalised Literature Review” requires an extensive revision prior the publication in Toxin journal.

The manuscript requires an overall revision in term of typographical style.

The text is pure informative and not improve knowledge on this topic.

The tables are difficult to read and refers limited amount of information. Authors should consider the opportunity of re-organize the text in the tables improving the information reported.

Following only few punctual suggestions:

-The abstract. The paragraph lines 12-17 should be rewritten

-Line 27. Botulinum neurotoxins are produced by several Clostridia species. Authors should mention all the species capable of producing botulinum toxins.

-Line 54. BT or BNT. Authors should define and use the same acronym.

-Line 76. Authors should specify the significance of the acronyms reported in the sentence.

Author Response

Many thanks for your advice on our manuscript.

Reviewer 3

The manuscript entitled “Lantox – The Chinese Botulinum Toxin Drug. Complete English Bibliography and Formalised Literature Review” requires an extensive revision prior the publication in Toxin journal.

Issue #1: The manuscript requires an overall revision in term of typographical style.

Reply: We are puzzled. What is typographical style? We used Times New Roman as advised. Chief editor, pls advise.

Issue #2: The text is pure informative and not improve knowledge on this topic.

Reply: Again, we are puzzled. Is the text informative or does it not improve knowledge on this topic?

Issue #3: The tables are difficult to read and refers limited amount of information. Authors should consider the opportunity of re-organize the text in the tables improving the information reported.

Reply: Re-organize the information in what way? Pls advise.

Following only few punctual suggestions:

Issue #1: The abstract. The paragraph lines 12-17 should be rewritten

Reply: This is the objected passage. What exactly should be re-written and how? We would be happy to do so. (4) and surgical (1). 73 publications are randomised controlled trials, 42 interventional studies, 235 observational studies, 10 case studies, 18 reviews and 2 guidelines. LAN publications cover all relevant topics of BT therapy throughout a period of more than 20 years. This constitutes a publication basis similar to those of other BT drugs. None of the publications analysed presented data contradictory to those generated with other BT drugs. LAN seems to have similar efficacy and safety features when compared to onabotulinumtoxinA using a 1:1 conversion ratio.

Issue # 2: Line 27. Botulinum neurotoxins are produced by several Clostridia species. Authors should mention all the species capable of producing botulinum toxins.

Reply: Botulinum toxin is exclusively produced by Clostridium botulinum. No other Clostridial species are producing botulinum toxin.

Issue #3: Line 54. BT or BNT. Authors should define and use the same acronym.

Reply: BT is defined as botulinum toxin consisting of botulinum neurotoxin and complexing proteins. BNT is defined as botulinum neurotoxin only.

Issue #4: Line 76. Authors should specify the significance of the acronyms reported in the sentence.

Reply: As requested by the reviewer the passage now reads: LAN's CP include the hemagglutinin (HA) components HA70 (57kDa, 17kDa), HA33 (30kDa, 28kDa) and HA17 (15kDa) and the non-toxic-non-hemagglutinin component NTNH (136kDa) (Liang et al. 2017).

Reviewer 4 Report

 Dear authors;

The article is a very valuable,  not only for neurological literature, but also for other fields of medicine. The subject of the article is very interesting and related to the common neurological problem. The article was written in a correct and comprehensive language, the English is understandable, and the results of the literature search provide an advance in current knowledge, including the history of working with botulinum toxin. The data and analyses are presented appropriately. In my opinion the highest standards for presentation of the results are used and the conclusions are interesting for the readership of the Journal. Additionally the work provide an advance towards the current knowledge. The results of the literature  search were presented correctly and clearly. The literature review is very rich and closely related to the subject of the article. The tables in the article help to understand the complexity  of the topic, which was taken under consideration.

I suggest a small changes in the order of application of the BT with the field of bruxism.
References should be corrected.

Author Response

Many thanks for your work on our manuscript.

Reviewer 4

The article is a very valuable, not only for neurological literature, but also for other fields of medicine. The subject of the article is very interesting and related to the common neurological problem. The article was written in a correct and comprehensive language, the English is understandable, and the results of the literature search provide an advance in current knowledge, including the history of working with botulinum toxin. The data and analyses are presented appropriately. In my opinion the highest standards for presentation of the results are used and the conclusions are interesting for the readership of the Journal. Additionally the work provide an advance towards the current knowledge. The results of the literature search were presented correctly and clearly. The literature review is very rich and closely related to the subject of the article. The tables in the article help to understand the complexity of the topic, which was taken under consideration.

Issue #1: I suggest a small changes in the order of application of the BT with the field of bruxism.

Reply: LAN usage in bruxism is limited. It may have been used in few patients with oromandibular dystonia. Also, some patients with TMJ disorder may actually have suffered from bruxism.

Round 2

Reviewer 1 Report

The major concerns of the first review have not been addressed in this revision. 

The article itself remains inconsistent in the presentation and intermixes 2 aspects: 1) providing information on the nature and subject matter of publications on Lantox and 2) summarizing the results of studies in the Lantox literature.

The summary of results is unreliable, as it is based on "selected" articles without stated selection criteria.  There is no information on the objective quality of the cited publication beyond whether they were RCTs... for example, whether the study was blinded, whether it used validated outcome measures, whether it was properly powered and complete information on the incidence of adverse events.  Without such information, there is no way to know if the stated conclusion that Lantox is of identical efficacy and safety to onabotulinumtoxinA is true or valid.  

It is not clear that this major deficiency could be addressed though further revision.

Some of the other minor issues identified during the first review have been at least partly addressed. 

Author Response

Many thanks for your work on our ms. We have revised it according to your advice. Please see the attachment.

Reviewer 3 Report

The revised manuscript included all reviewers' comments and in this respect can be considered worthy for publication.

Author Response

Thank you for your work and comment on our manuscript. We replied the reviewer's comments and revised our manuscript according to their advice.